# Analysis of Hand Function, Upper Limb Disability, and Its Relationship with Peripheral Vascular Alterations in Raynaud’s Phenomenon

**DOI:** 10.3390/diagnostics14010093

**Published:** 2023-12-30

**Authors:** Rosa Mª Tapia-Haro, Mª Carmen García-Ríos, Adelaida Mª Castro-Sánchez, Sonia Toledano-Moreno, Antonio Casas-Barragán, Mª Encarnación Aguilar-Ferrándiz

**Affiliations:** 1Department of Physical Therapy, Faculty of Health Sciences, University of Granada (UGR), 18071 Granada, Spain; rtapia@ugr.es (R.M.T.-H.); mcgrios@ugr.es (M.C.G.-R.); antoniocb@ugr.es (A.C.-B.); 2ibs.GRANADA Instituto de Investigación Biosanitaria, 18012 Granada, Spain; sonia10toledano@gmail.com; 3Department of Nursing, Physical Therapy and Medicine, University of Almeria, 04120 Almeria, Spain; adelaid@ual.es; 4Biomedicine Program, Department of Physical Therapy, Faculty of Health Science, University of Granada (UGR), 18071 Granada, Spain

**Keywords:** Raynaud disease, systemic scleroderma, upper extremity, hand, disability evaluation, activities of daily living

## Abstract

This study aimed to compare vascular involvement, hand functionality, and upper limb disability between Raynaud’s phenomenon participants and controls. Also, we analyzed the relationships between vascular impairment, mobility, and strength with disability in this Raynaud population. We conducted a case–control study with fifty-seven participants. We registered sociodemographic and clinical data; vascular variables (temperature, cold test, blood flow, and oxygen saturation); functional variables (pinch strength, range of motion), and disability (Shortened Disabilities of the Arm, Shoulder and Hand Questionnaire) (Q-DASH). Raynaud participants present more disability in all Q-DASH subscales, lower hands’ temperature pre and post cold test, decreased blood flow on radial artery, decreased ranges of motions at passive extension of index finger, and active flexion and extension of thumb than the healthy controls. The multivariate regression analysis showed that extension of the index finger, lateral pinch strength, and oxygen saturation were significantly associated with disability in RP, almost the 55% of the total variance on the upper limb, 27% at sports/arts, and 42% at work. Our findings suggest that RP has a disabling effect on the upper extremities and a practice of activities in people who suffer it. Also, disability in Raynaud seems to be more related with hand mobility and strength impairment than vascular injury.

## 1. Introduction

Raynaud’s phenomenon (RP) is a complex of symptoms [1,2] defined as the clinical expression of a vasospastic disorder, characterized by an acute and transient decrease in blood flow on the upper and lower extremities in response to exposure to cold or emotional stressors [2]. Attacks mainly involve the fingers of the hands; toes are also involved in up to 40% of patients [3,4]. RP is a common disorder in the general population with a prevalence ranging between 3 and 21%. RP affects women more frequently than men with a female/male ratio of 7:1 [5,6].

RP is classified as primary (PRP) or idiopathic and secondary (SRP). PRP is the most habitual with 80% of cases [3]. The secondary form of RP is associated with an underlying disease, usually autoimmune diseases and connective tissue disorders. Within these disorders, PR is a characteristic early-onset cutaneous manifestation that occurs in patients with systemic scleroderma [7] and appears in up to 90 % of these patients, between 10 and 45 % with systemic lupus erythematosus, in 33 % with Sjögren’s syndrome, and in 20 % with dermatomyositis/polymyositis [8].

The main symptom of RP is a characteristic triphasic color change in the skin: white (vasospasm/ischaemia); blue (deoxygenation/cyanosis); and red (hyperemia) [6]. Other symptoms include a variable degree of pain, paresthesia, numbness, tingling, and open sores on the digits [6]. Attacks usually last between ten and twenty minutes [9]. All of these symptoms cause loss of hand function, distress, and reduced quality of life in RP patients [10,11]. Disability occurs in both forms of RP and depends on multiple factors, including duration, frequency, and severity of attacks; the presence of ulcers on the fingers; pain; and concomitant pathologies [12]. Most patients with RP (71% with PRP and 87% with SRP) [13] report having to change their daily routine because of functional disability and significant levels of anxiety [14]. Although RP is not a potentially life-threatening condition, it has a considerable impact on the general state of health and quality of life and becomes a socioeconomic and emotional burden for these patients [3,8,15]. However, the literature that analyze hand and upper limb disability in RP patients and its effects on the activities of daily living (ADLs) is limited.

The European Society of Vascular Medicine (ESVM) [8] established basic recommendations for the diagnosis of PR, such as making a complete medical history and a complete blood count. A three-step protocol for the diagnosis of RP and five additional criteria for the diagnosis of primary RP have also been determined [1]. The scientific literature proposes different diagnostic tools for RP [4,6] from self-reports, such as the daily Raynaud’s Condition Score (RCS), measures of the duration, and frequency of RP attacks, to a general physical assessment and a specific evaluation of the hand [6,9]. Different non-invasive techniques have also been determined to evaluate anomalies at the vascular level, such as nail capillaroscopy, thermography, and Doppler lasers [9]. Furthermore, the functional response to cold is usually evaluated by performing a localized cold test together with an analysis of the biochemical profile, to complete the diagnosis [9]. In recent years, the need to consider measures of the severity and impact of episodes has been highlighted, such as an assessment of pain using the Visual Analog Scale (VAS) and an assessment of disability in activities of daily living in patients with RP [8,9]. In this regard, a complete exploration of the musculoskeletal aspects that enable the functionality of the hand with parameters such as joint amplitude ranges and muscle strength should be considered. The disability caused by RP must also be measured in the different areas of daily life of these patients. However, these assessment measures have not been specified for this condition [1,8,9].

With this background, we consider that it is necessary to explore in depth, with simple and non-invasive tools, the relationship between vascular impairment and functionality with upper limb disability in subjects with primary and secondary RP. We hypothesize that RP patients should have higher levels of disability in the upper limbs and therefore will have more difficulty carrying out ADLs and activities related to work, sports, and arts than the healthy controls because of vascular, mobility, and strength impairments.

Thus, the aim of this study was to evaluate the vascular involvement, hand functionality, and disability in the upper extremities in participants with RP in comparison with a healthy control group. Secondly, we attempted to investigate if there were relationships between vascular variables (temperature, cold test, oxygen saturation, and blood flow), functionality variables (range of motion and strength), and disability in participants with Raynaud.

## 2. Materials and Methods

### 2.1. Study Design and Participants

This was a pilot case–control study. A total of 57 participants were included in this study. RP participants were recruited from the Rheumatology Service of the Virgen de las Nieves Hospital in Granada (Spain) and the controls from volunteers who responded to a local advertisement by the Physiotherapy Department of the University of Granada. The inclusion criteria were (a) age over 18 years; (b) previous diagnosis of PR (primary or secondary), according to the criteria established by LeRoy and Medsger [16]; and (c) a history of at least one year of regular attacks of RP. The exclusion criteria were (a) the presence of skin alterations (scars, ulcers, gangrene, or bites in the area to be examined); (b) the presence of upper limb entrapment syndrome, polyneuropathy, or renal failure; (c) pregnant or lactating women; (d) the use of vibratory tools; (e) a history of drug or alcohol abuse; and (f) the presence of a tumoral process.

### 2.2. Procedures

Informed consent forms were signed by all the participants of the study, which was approved by the Bioethics Committee of the University of Granada (Spain) on 12 May 2015 (No. 27/CEIH/2015) and conducted in accordance with the amended version of the Declaration of Helsinki, 2013.

### 2.3. Measures

Each participant was evaluated over a total time of 90 min on the same day. Firstly, participants were asked for their sociodemographic and clinical data, such as age, sex, hand dominance, current pharmacologic treatment, number of RP attacks per week, presence of comorbidities and disability by using the Shortened Disabilities of the Arm, Shoulder and Hand Questionnaire (Quick-DASH). Then we measured blood flow, oxygen saturation, pinch strength, and range of motion. Finally, temperature and cold test assessments were carried out.

#### 2.3.1. Vascular Assessment

##### Temperature Assessment and Cold Stress Test (CST)

A hand-held infrared thermographic scanner (Derma Temp^®^, Model: 104920-DT-1001-LT, Exergen Corporation, Watertown, MA, USA) was used to measure the temperature in degrees Celsius (°C), in the fingertip of the third finger on both hands of the participants. Three parameters were obtained: pre-CST, post-CST, and recovery temperature. The pre-CST temperature was obtained after acclimatization [17,18]. The CST was used to evaluate vascular response to changes in temperature following the protocol described in previous studies, where both hands were immersed for 2 min in cool water at 10 °C [17,18]. After the CST, the temperature was taken to obtain the post-CST. Finally, the recovery temperature was obtained by subtracting the basal temperature minus the final temperature after 45 min of the CST.

##### Oxygen Saturation (SpO2)

A finger pulse oximeter (MEGOS Oxi-Pulse^®^, SONMEDICA S.A, [Barcelona, Spain]) was used to determine the percentage of SpO2 [19]. The oximeter was placed on the middle finger of both hands. Pulse oximetry is a quick and well-established test used to quantify SpO2, and devices available today are very reliable [19,20].

##### Arterial Blood Flow

Blood flow in the radial and ulnar artery was evaluated with a Hadeco Bi-Directional Vascular Doppler^®^ (Hadeco Inc., Arima, Miyamae-ku Kawasaki, Japan) following the protocol described by Toprak et al. [21]. The radial and ulnar blood flows were evaluated on the volar surface of the wrists of both hands and the mean of three measurements was calculated. The results were expressed in cm/s^−2^ [21].

#### 2.3.2. Functionality Assessment

##### Quick-DASH Questionnaire

Upper limb disability was measured with the Spanish version of the Q-DASH [22]. This is a standardized self-administered questionnaire that assesses the patient’s perceived disability to perform activities [23]. The questionnaire has 11 items, with a 5-point Likert scale range. Q-DASH has two optional additional modules with 4 items (Work and Sports/Performing Arts). The scores range from 0 (no disability) to 100 (most severe disability), where higher scores indicate a greater level of disability [23]. This questionnaire has demonstrated good reliability, validity, and responsiveness, with a Cronbach alpha of 0.90 and a test–retest Pearson correlation coefficient of 0.70 [23].

#### Range of Motion (ROM) in Index Finger and Thumb

ROM was assessed with a stainless-steel finger goniometer (SAHEAN^®^) following a protocol based on scientific evidence [24,25,26]. The ROM of active and passive flexion and extension of the metacarpophalangeal joint, flexion of the proximal interphalangeal joint, and flexion of the distal interphalangeal joint in the index (second) finger was evaluated in both hands. Active and passive flexions were evaluated in the thumb at the interphalangeal joint, as well as flexion and extension at the metacarpophalangeal joint [25]. Each outcome was recorded three times following the same sequence of angles, and the mean of the three values was used in the analysis [27]. Goniometer measurements have a high inter-rater reliability and are considered a valid tool [27].

##### Pinch Strength

A mechanical finger dynamometer Pinch Gauge (Baseline^®^) was used to assess pinch strength [28]. Two types of pinch strength between the thumb and index fingers were measured for both hands: the tip and lateral pinch strength [29]. Standardized positioning was used in accordance with recommendations by the American Society of Hand Therapists [30]. Participants were instructed to compress the dynamometer as hard as possible with the fingers. The final score for each pinch was defined by calculating the mean of the three values measured [29]. The Pinch Gauge has demonstrated high inter-rater and test–retest reliability, with an intraclass correlation coefficient higher than 0.90 [28,29,31].

### 2.4. Statistical Analysis

We used the Ene 3.0 software (Autonomous University of Barcelona, Barcelona, Spain) to calculate the sample size, using the data from the pilot study phase of this project, which included a total of 18 participants. For Q-DASH, the power analysis revealed that 14 patients were necessary in each group to obtain a desired power (β) of 90% with a significance level α = 0.05 and to allow 50% of losses. The SPSS© version 20.0 (IBM Corporation, Armonk, NY, USA) was used for data analysis. Firstly, normality of the variables (*p* > 0.05) was evaluated using the Kolmogorov–Smirnov test. Demographics and clinical variables between groups were compared using ANOVA for continuous data and χ^2^ for categorical data. In order to assess the main objective of the study, between-groups differences were evaluated via an ANCOVA analysis in which the key variables (vascular and functional outcomes) were the between-subjects factor, and the covariates were age and gender. Post hoc analyses were carried out using Bonferroni correction for multiple comparisons. Since there were no differences between dominant and non-dominant sides, the unified average value of both hands was calculated for key variables. A Pearson bivariate correlation analysis was subsequently performed to evaluate the relationship between vascular, functional variables and Q-DASH in RP groups. Finally, a multivariate regression analysis was carried out. After the collinearity analysis, index extension, thumb flexion, oxygen saturation, and lateral pinch strength were included as independent variables, and Q-DASH (and its subscales) as the dependent variable. All analyses were two-tailed and *p* < 0.05 was considered significant.

## 3. Results

### 3.1. Sociodemographic and Clinical Characteristics

A total sample of 57 subjects with a mean (SD) age of 41.7 (15.5) years was recruited for this study: eighteen with PRP (72.2% females), nineteen with SRP to systemic scleroderma (78.9% females), and twenty healthy controls (80% females) (Table 1).

Regarding the pharmacological treatment of the participants in the study, it consisted mainly of analgesic drugs 21 (36.8%); non-steroidal anti-inflammatory drugs 16 (28.1%), vasodilators 8 (14%), antidepressants 7 (12.3%), and insulin 2 (3.5%).

### 3.2. Vascular Assessment

ANOVA analysis showed significant differences between groups in the finger temperature for both hands at pre-CST (F = 4.04, *p* = 0.023) and post-CST (F = 8.22, *p* = 0.001). Significant differences between groups were only achieved at recovery temperature for the non-dominant side (NDS): F = 3.28, *p* = 0.045. Also, it showed statistically significant differences between groups for blood flow in the radial artery [dominant side (DS): F = 5.24, *p* = 0.008; NDS: F = 8.29, *p* = 0.001]. There were no differences for the rest of the variables (F ≥ 0.76, *p* ≥ 0.360). Mean and standard deviation are shown in Table 1.

Post hoc analysis showed that the pre-CST temperature was significantly lower in the PRP in comparison to the controls. Similarity, post-CST temperature was significantly lower in the PRP group in comparison to the controls and the SRP group. Recovery temperature was significantly higher in PRP in comparison to the controls. Also, Primary and Secondary Raynaud participants had significantly lower blood flow in the radial artery than the controls. There were no differences between primary and secondary Raynaud and the controls in ulnar artery blood flow and oxygen saturation (Table 2).

### 3.3. Functionality Assessment

ANOVA analysis showed statistically significant differences between groups in the Q-DASH (Total: F = 81.56, *p <* 0.001; Work: F = 56.46, *p <* 0.001; Sports/Arts: F = 54.16, *p <* 0.001); in range of motion for passive index finger extension (DS: F = 3.593, *p* = 0.034; NDS: F = 3.233, *p* = 0.047); active thumb flexion (DS: F = 3.540, *p* = 0.036; NDS side: F = 3.449, *p* = 0.039); and active thumb extension (NDS: F = 6.87, *p* = 0.002). There were no differences for the rest of the functionality variables (F ≥ 0.07, *p* ≥ 0.063) (Table 1).

Post hoc analysis revealed that patients with Raynaud showed significantly higher upper limb disability and higher disability for Work and Sports/Arts subscales than the healthy controls. Furthermore, SRP group showed significantly higher disability for the three Q-DASH subscales in comparison with the PRP group. Patients with PRP had a significantly higher range of motion than the controls for passive flexion and passive extension of the index finger, and SRP participants for active flexion and active extension of the thumb (Table 2).

### 3.4. Vascular and Functional Factors Associated with Disability in RP Participants

Bivariate correlation analysis showed that flexion and extension of the index finger, flexion and extension of the thumb, and lateral pinch strength were statistically, indirectly associated with general upper limb disability and disability related to work/sports/arts activities in RP subjects (0.230 ≥ r ≤ 0.613, 0.000 ≥ *p* ≤ 0.032). In addition, lower levels of oxygen saturation and temperature post-CST were also significant and were associated with higher scores in work disability (Table 3).

### 3.5. Final Multiple Regression Model of Predictive Factors Associated with Disability in RP Participants

Multivariate regression analysis showed that extension of the index finger and lateral pinch strength were significantly associated with the dependent variable Q-DASH, predicting almost 55% of the total variance on upper limb disability in RP patients (Table 4). Similar results were achieved when Sports/Arts disability was used as the dependent variable, with both independent variables explaining almost 27% of the total variance (Table 5). When disability in the work subscale was taken into account as a dependent variable, the multivariate model showed that oxygen saturation and extension of the index finger were significantly associated with the dependent variable, predicting almost 42% of the total variance (Table 6).

## 4. Discussion

In this study, subjects with RP had lower hand temperature at the baseline and after the CST, less blood flow on the radial artery, less ROM for passive extension of the index finger, and active flexion and extension of the thumb than the controls. They also showed more disability in all subscales of the Q-DASH. The multivariate regression analysis confirmed that index finger extension, lateral pinch strength, and oxygen saturation were significantly associated with disability in RP participants. These results are in line with the hypothesis of the current pilot study; however, disability in RP participants seems to be more related to ROMs’ impairment and hand strength than the severity of vascular alterations.

With respect to temperature, the current results were consistent with previous studies [32,33,34] that showed similar findings for temperature records before and after CST and recovery patterns in PRP and SRP participants and the healthy controls. The blood flow measurements were also consistent with previous studies [35,36,37,38] that reported lower baseline blood flow in RP patients than in the controls. Furthermore, these authors [35,38] found no significant difference in the basal blood flow between the PRP and SRP groups. With regard to oxygen saturation, our results coincide with those of a recent study [39] on SRP patients, which determined that there was no statistically significant difference between the groups at baseline.

According to our knowledge, there are no previous studies that include the ROMs’ assessment and the relationship between ROMs and disability in RP patients. Some studies in rheumatoid patients show that a decreased ROM in the joints is related to disabilities of the arm, shoulder, and hand [28,31,40]. Bain et al. [24] mentioned in their study that patients with loss of finger joint extension have difficulties forming a grip. This relates to our results, where index extension is a predictive factor associated with disability in the three Q-DASH subscales in RP participants in the final multiple regression analysis.

With respect to pinch strength, different studies [14,41,42] of the reviewed literature agree that RP associated with scleroderma has a severe debilitating effect on patients. This supports our results about tip and lateral pinch strength measurements that were lower in the SRP group.

We have found several studies on systemic sclerosis [11,39,41,42,43], rheumatic pathologies [40], and hand–arm vibration syndrome [44] which agree that disability rates are significantly higher in patients with associated RP compared to those without RP. The loss of hand function occurs in both RP groups. Disability rates measured using the Q-DASH were significantly higher among SRP patients than PRP participants. Along these lines, a previous study [3] reported that functionality was influenced by age and comorbidities; an older age may be associated with more comorbidities, which may be related to worse Q-DASH scores. A study on patients with Systemic Sclerosis [45] reported that the disease had a disabling effect on ADLs and work. Another study [46] also mentioned that daily activities are less affected in PRP patients than in those with the secondary form of Raynaud.

It can be interpreted from our findings that disability in RP patients depends on multiple factors and that the relationship between vascular impairment and activity limitations is not evident. Disability of the upper limbs, work, and sport/arts in RP patients is related more to loss of hand mobility and pinch strength grip. Previous studies along these lines mentioned [44,47] that upper limb disability in RP patients is related to the frequency of blanching attacks, but this is not reflected in our results. Mason et al. [44] suggested that in patients with SRP and hand–arm vibration syndrome, upper limb disability is related more to sensorineural components than to vascular symptoms. A recent study [11] in patients with SRP and early systemic sclerosis concluded that the number of RP attacks and the difficulties associated with them were linked to limitations in all ADL domains.

The exact way in which RP affects hand functionality remains unclear. The functionality of the hand can be compromised in these patients not only due to vascular alterations, but also due to other aspects to consider, such as thickening of the skin, as well as pain and inflammation characteristic of the pathologies to which it may be associated. We believe that additional studies are necessary, including measures that assess these aspects, since this could help improve knowledge about RP, as well as determine the burden of the disease (psychological, socioeconomic) and the impact of RP on the activities of the daily life and the quality of life of people who suffer from it.

This study has several strengths, such as our data support the concept that RP is a complex clinical condition that has a significant impact on the general health status of people who suffer it. To the best of our knowledge, this study represents the first attempt to explore the relationship between vascular and functional impairment and disability in different activity domains (ADLs, work, sports, and arts) in subjects with primary and secondary RP.

Our study has several limitations. Firstly, although it has an adequately powered sample size to detect differences between RP participants and controls, a larger sample size would be needed in future studies to extrapolate the data. Secondly, two heterogeneous age groups of RP participants were studied. However, so as not to alter the results obtained, age was included as a covariate in the ANCOVA analysis and added to the regression model as a continuous variable following the protocol mentioned in a previous study [48]. Thirdly, we have only included in our study patients with SRP due to systemic scleroderma with mild involvement, since we excluded patients who presented scars, ulcers, gangrene, or bites in the area to be examined, so future studies that include patients with SRP due to other pathologies and in different phases of involvement are necessary. Fourthly, this was an observational study and we cannot be certain that the differences observed were not due to differences between groups related to diverse factors, such as medical comorbidities. Finally, in the case of the Q-DASH, we chose an instrument that was validated for hand and upper extremity conditions, but was not specifically for RP.

## 5. Conclusions

In conclusion, the exploration that we have performed suggest that upper limb disability in the RP participants limits the practice of ADLs, work, sports, and arts, especially when Raynaud is associated with an underlying disease. Disability in the RP seems to be related more to loss of ROMs and strength than to vascular alterations. The present findings should provide valuable information for future studies to improve the diagnosis and treatment of this pathology.

## Figures and Tables

**Table 1 diagnostics-14-00093-t001:** Sociodemographic, vascular, and functionality characteristics of participants.

Outcomes	PRP ^†^ *n* = 18	SRP ^‡^ *n* = 19	Controls *n* = 20
Age in years, mean (standard deviation)	28.4 (10.4)	55.8 (6.2)	40.3 (14.3)
Sex, *n* (%)			
Male	5/27.8	4/21.1	4/20
Female	13/72.2	15/78.9	16/80
Hand dominance (%)			
Right	18/100	19/100	18/90
Left	-	-	2/10
RP attacks (No./week) (%)	23.3 (7.2)	28.0 (21.0)	-
Associated Pathologies (%)			
Arterial Hypertension	-	6/31.6	2/10
Hypercholesterolemia	-	2/10.5	1/5
Diabetes	-	2/10.5	-
Vascular Items, mean (standard deviation)			
Temperature pre-CST ^§^ (°C)			
Dominant	26.3 (3.8)	29.0 (4.3)	29.8 (3.9)
Non-Dominant	26.6 (4.0)	29.4 (4.0)	30.1 (3.8)
Both hands	26.5 (3.9)	29.2 (4.1)	30.0 (3.9)
Temperature post-CST (°C)			
Dominant	24.4 (4.4)	27.7 (5.2)	30.3 (3.7)
Non-Dominant	24.1 (4.7)	27.8 (4.9)	29.9 (4.0)
Both hands	24.2 (4.6)	27.8 (5.0)	30.1 (3.8)
Recovery Temperature (°C)			
Dominant	1.8 (3.2)	1.3 (3.5)	−0.4 (2.9)
Non-Dominant	2.6 (3.3)	1.6 (3.1)	0.2 (2.4)
Both hands	2.2 (3.2)	1.5 (3.2)	−0.1 (2.6)
Blood flow radial artery (cm/s^−2^)			
Dominant	8.5 (2.7)	8.6 (4.8)	12.4 (5.0)
Non-Dominant	8.5 (2.2)	8.8 (3.6)	12.5 (3.9)
Blood flow ulnar artery (cm/s^−2^)			
Dominant	10.0 (3.6)	10.9 (6.8)	11.8 (3.4)
Non-Dominant	8.3 (2.3)	15.0 (24.0)	12.4 (4.2)
Oxygen Saturation (%)			
Dominant	97.5 (0.9)	96.6 (1.1)	97.1 (1.2)
Non-Dominant	97.6 (0.8)	96.9 (0.9)	97.0 (1.3)
Functionality Items, mean (standard deviation)			
Quick-DASH ^||^ (%)			
Upper limb disability	15.9 (11.4)	57.3 (13.9)	2.5 (7.6)
Work Module	21.9 (19.1)	72.7 (24.6)	1.9 (8.4)
Sports/Performing Arts Module	22.9 (25.5)	69.7 (19.8)	0 (0.0)
ROM ^¶^ Active index finger flexion			
Dominant	93.2 (6.2)	81.5 (9.9)	84.8 (8.9)
Non-Dominant	93.2 (6.8)	81.8 (9.4)	85.7 (9.1)
ROM Passive index finger flexion			
Dominant	103.2 (5.3)	93.3 (8.9)	95.7 (8.0)
Non-Dominant	104.2 (5.5)	93.9 (7.6)	96.0 (7.7)
ROM Active index finger extension			
Dominant	32.5 (6.0)	30.5 (6.6)	30.6 (6.1)
Non-Dominant	31.8 (5.6)	30.8 (7.5)	32.0 (5.6)
ROM Passive index finger extension			
Dominant	60.8 (9.7)	49.7 (12.8)	49.5 (14.4)
Non-Dominant	61.9 (9.1)	49.0 (12.5)	52.1 (12.3)
ROM Active thumb flexion			
Dominant	76.2 (4.4)	67.2 (7.4)	72.2 (7.2)
Non-Dominant	78.3 (5.4)	69.9 (9.6)	73.4 (7.5)
ROM Passive thumb flexion			
Dominant	87 (5)	72.1 (18.8)	82.3 (8.9)
Non-Dominant	87.9 (6.2)	79.2 (9.9)	84.4 (7.8)
ROM Active thumb extension			
Dominant	30.2 (5.5)	23.4 (5.5)	25.5 (6.5)
Non-Dominant	29.4 (3.4)	21.6 (6.5)	25.6 (5.6)
ROM Passive thumb extension			
Dominant	51.9 (8.9)	43.9 (9.5)	46.6 (11.3)
Non-Dominant	51.9 (8.6)	44.2 (8.7)	46.5 (12.9)
Tip pinch strength			
Dominant	4.8 (1.6)	4.7 (2.4)	5.1 (1.5)
Non-Dominant	4.5 (1.8)	4.5 (2.4)	4.9 (1.4)
Lateral pinch strength			
Dominant	7.5 (1.6)	6.4 (2.0)	7.4 (2.5)
Non-Dominant	7.6 (1.7)	6.1 (2.0)	7.1 (2.6)

Data are expressed as the mean and standard deviation (SD) for quantitative variables or frequency and % for qualitative outcomes. ^†^ PRP: Primary Raynaud’s phenomenon; ^‡^ SRP: Secondary Raynaud’s phenomenon; ^§^ CST: Cold Stress Test; ^||^ Quick-DASH: Shortened Disability of the Arm, Shoulder and Hand Questionnaire; ^¶^ ROM: range of motion.

**Table 2 diagnostics-14-00093-t002:** Mean Difference (MD), 95% Confidence Interval (CI) and between-groups’ level of significance for vascular and functionality assessments.

Outcomes	Controls vs. Primary RP ^†^	*p*-Value	Controls vs. Secondary RP ^†^	*p*-Value	Primary vs. Secondary RP ^†^	*p*-Value
	MD (95% CI)		MD (95% CI)		MD (95% CI)	
Vascular Items						
Temperature pre-CST ^‡^ (°C)						
Dominant	3.540 (0.307; 6.772)	0.027 *	0.818 (−2.368; 4.006)	1.000	−2.721 (−5.994; 0.551)	0.134
Non-Dominant	3.440 (0.288; 6.591)	0.028 *	0.626 (−2.481; 3.734)	1.000	−2.813 (−6.004; 0.377)	0.101
Both hands	3.490 (−0.317; 6.662)	0.026 *	0.722 (−2.405; 3.851)	1.000	−2.767 (−5.978; 0.444)	0.114
Temperature post-CST (°C)						
Dominant	5.918 (2.320; 9.517)	0.000 *	2.569 (−0.979; 6.118)	0.238	−3.349 (−6.992; 0.293)	0.081
Non-Dominant	5.848 (2.193; 9.503)	0.001 *	2.104 (−1.499; 5.708)	0.465	−3.743 (−7.444; −0.043)	0.046 *
Both hands	5.883 (2.285; 9.482)	0.001 *	2.336 (−1.211; 5.885)	0.328	−3.546 (−7.189; 0.096)	0.059
Recovery Temperature (°C)						
Dominant	−2.267 (−4.844; 0.308)	0.102	−1.750 (−4.291; 0.790)	0.283	0.517 (−2.091; 3.125)	1.000
Non-Dominant	−2.408 (−4.756; −0.060)	0.043 *	−1.451 (−3.766; 0.863)	0.382	0.957 (−1.419; 3.334)	0.973
Both hands	−2.338 (−4.741; 0.065)	0.059	−1.600 (−3.971; 0.769)	0.303	0.737 (−1.690; 3.170)	1.000
Blood flow radial artery (cm;s^−2^)						
Dominant	4.522 (0.707; 8.338)	0.015 *	2.954 (−1.050; 6.957)	0.221	−1.569 (−6.644; 3.507)	1.000
Non-Dominant	3.973 (0.975; 6.970)	0.006 *	3.694 (0.549; 6.840)	0.016 *	−0.279 (−4.266; 3.709)	1.000
Blood flow ulnar artery (cm;s^−2^)						
Dominant	2.155 (−2.180; 6.490)	0.674	0.423 (−4.126; 4.972)	1.000	−1.732 (−7.499; 4.035)	1.000
Non-Dominant	5.408 (−7.147; 17.963)	0.875	−4.222 (−17.396; 8.952)	1.000	−9.630 (−26.331; 7.071)	0.480
Oxygen Saturation (%)						
Dominant	−0.129 (−1.064; 0.807)	1.000	0.281 (−0.700; 1.262)	1.000	0.410 (−0.834; 1.654)	1.000
Non-Dominant	−0.262 (1.141; 0.617)	1.000	−0.353 (−1.275; 0.570)	1.000	−0.091 (−1.261; 1.078)	1.000
Functionality Items						
Quick-DASH ^§^ (%)						
Upper limb disability	−14.330 (−24.294; −4.367)	0.002 *	−53.596 (−64.051; −43.141)	0.000 *	−39.266 (−52.520; −26.012)	0.000 *
Work Module	−17.982 (−34.354; −1.610)	0.027 *	−73.468 (−56.289; −90.647)	0.000 *	−55.486 (−33.707; 77.265)	0.000 *
Sports/Performing Arts Module	−21.885 (−38.203; −5.568)	0.005 *	−71.089 (−88.211; −53.976)	0.000 *	−49.203 (−70.910; 27.497)	0.000 *
ROM ^||^ Active index finger flexion						
Dominant	−6.612 (−14.110; 0.887)	0.101	−0.989 (−6.880; 8.856)	1.000	7.600 (−2.375; 17.575)	0.195
Non-Dominant	−5.106 (−12.455; 2.244)	0.275	0.728 (−6.983; 8.440)	1.000	5.834 (−3.942; 15.610)	0.438
ROM Passive index finger flexion						
Dominant	−6.095 (−12.780; 0.591)	0.085	0.506 (−6.509; 7.521)	1.000	6.600 (−2.293; 15.493)	0.216
Non-Dominant	−6.099 (−12.160; −0.038)	0.048*	−0.619 (−6.979; 5.740)	1.000	5.480 (−2.583; 13.542)	0.296
ROM Active index finger extension						
Dominant	−2.080 (−7.655; 3.495)	1.000	0.310 (−5.540; 6.160)	1.000	2.390 (−5.026; 9.806)	1.000
Non-Dominant	0.571 (−5.310; 6.173)	1.000	0.753 (−5.125; 6.631)	1.000	0.182 (−7.269; 7.634)	1.000
ROM Passive index finger extension						
Dominant	−11.871 (−23.052; −0.689)	0.034 *	0.583 (−11.150; 12.316)	1.000	12.453 (−2.420; 27.327)	0.130
Non-Dominant	−9.583 (−19.819; 0.652)	0.074	2.926 (−7.814; 13.666)	1.000	12.510 (−1.106; 26.125)	0.082
ROM Active thumb flexion						
Dominant	−3.614 (−9.430; 2.201)	0.391	4.696 (−1.406; 10.798)	0.188	8.310 (0.574; 16.046)	0.031 *
Non-Dominant	−5.451 (−12.33; 1.436)	0.167	4.152 (−3.074; 11.378)	0.484	9.603 (0.442; 18.764)	0.037 *
ROM Passive thumb flexion						
Dominant	−3.351 (−14.364; 7.662)	1.000	8.510 (−3.046; 20.066)	0.223	11.861 (−2.789; 26.511)	0.151
Non-Dominant	−2.236 (−9.403; 4.932)	1.000	3.626 (−3.895; 11.147)	0.716	5.862 (−3.673; 15.397)	0.430
ROM Active thumb extension						
Dominant	−4.475 (−9.713; 0.762)	0.118	1.828 (−3.668; 7.324)	1.000	6.0303 (−0.664; 13.271)	0.089
Non-Dominant	−4.538 (−9.286; 0.210)	0.065	4.930 (−0.051; 9.912)	0.053	9.468 (3.153; 15.784)	0.002 *
ROM Passive thumb extension						
Dominant	−4.790 (−13.719; 4.139)	0.571	2.041 (−7.328; 11.411)	1.000	6.832 (−5.046; 18.709)	0.483
Non-Dominant	−5.524 (−14.742; 3.693)	0.433	2.394 (−7.278; 12.066)	1.000	7.919 (−4.343; 20.180)	0.349
Tip pinch strength						
Dominant	0.546 (−1.519; 1.882)	1.000	0.546 (−1.238; 2.331)	1.000	0.365 (−1.897; 2.627)	1.000
Non-Dominant	0.462 (−1.258; 2.182)	1.000	0.359 (−1.446; 2.164)	0.730	−0.103 (−2.392; 2.185)	1.000
Lateral pinch strength						
Dominant	−0.011 (−1.884; 1.861)	1.000	0.805 (−1.160; 2.770)	0.947	0.817 (−1.674; 3.307)	0.817
Non-Dominant	−0.160 (−2.057; 1.736)	1.000	0.651 (−1.339; 2.641)	1.000	0.811(−1.711; 3.334)	1.000

* *p* < 0.05 for Bonferroni pairwise comparison between groups. ^†^ RP: Raynaud’s phenomenon; ^‡^ CST: Cold Stress Test; ^§^ Quick-DASH: Shortened Disability of the Arm, Shoulder and Hand Questionnaire; ^||^ ROM: range of motion.

**Table 3 diagnostics-14-00093-t003:** Bivariate correlations between Quick-DASH/vascular outcomes, range of motion, and strength in Raynaud’s phenomenon participants (*n* = 37).

Outcomes Measures	Quick-DASH ^†^ Upper Limb Disability		Quick-DASH ^†^ Work Module		Quick-DASH ^†^ Sports/Arts Module	
	Pearson (r)	*p*-Value	Pearson (r)	*p*-Value	Pearson (r)	*p*-Value
RP ^‡^ attacks (No./week)	0.141	0.406	0.648	0.078	0.412	0.139
Temperature pre-CST ^§^	0.219	0.193	0.254	0.129	−0.089	0.602
Temperature post-CST	0.293	0.079	−0.348	0.035 *	0.027	0.876
Recovery Temperature	−0.166	0.326	−0.213	0.205	−0.151	0.374
Blood flow radial artery	0.169	0.318	0.161	0.341	0.723	−0.060
Blood flow ulnar artery	0.284	0.089	0.239	0.153	0.111	0.512
Oxygen Saturation	−0.394	0.016 *	−0.535	0.001 **	−0.183	0.278
ROM ^||^ index finger flexion	−0.613	0.000 ***	−0.517	0.001 **	−0.522	0.001 **
ROM index finger extension	−0.525	0.001 **	−0.459	0.004 **	−0.401	0.014 *
ROM thumb flexion	−0.394	0.016 *	−0.408	0.012 *	−0.204	0.226
ROM thumb extension	−0.442	0.006 **	−0.473	0.003 **	−0.354	0.032 *
Tip pinch strength	−0.046	0.789	0.107	0.530	−0.079	0.642
Lateral pinch Strength	−0.459	0.004 **	−0.230	0.171	−0.389	0.017 *

* *p* < 0.05; ** *p* < 0.01; *** *p* < 0.001. ^†^ Quick-DASH: Shortened Disability of the Arm, Shoulder and Hand Questionnaire; ^‡^ RP: Raynaud’s phenomenon; ^§^ CST: Cold Stress Test; ^||^ ROM: range of motion.

**Table 4 diagnostics-14-00093-t004:** Final multiple regression model of predictive associated factors to upper limb disability in Raynaud’s phenomenon participants (*n* = 37).

Quick-DASH ^†^: Upper Limb Disability (r^2 ‡^ = 0.551)						
Independent Variables	B ^§^	95% CI ^||^		β ^¶^	SE ^#^	*p*-Value
		Upper Limit	Lower Limit			
Oxygen Saturation	−7.400	−0.016	−14.785	−0.273	3.625	0.050
ROM ^††^ index finger extension	−1.122	−0.291	−1.954	−0.348	0.408	0.010 *
ROM thumb flexion	−0.474	0.298	−1.246	−0.164	0.379	0.220
Lateral pinch strength	−5.706	−2.400	−9.012	−0.427	1.623	0.001 *

* *p* < 0.05; ^†^ Quick-DASH: Shortened Disability of the Arm, Shoulder and Hand questionnaire; ^‡^ r^2^: regression coefficient of determination; ^§^ B: regression coefficient; ^||^ CI: confidence interval; ^¶^ β: adjusted coefficient from multiple linear regression analysis; ^#^ SE: coefficient standard error; ^††^ ROM: range of motion.

**Table 5 diagnostics-14-00093-t005:** Final multiple regression model of predictive associated factors to sports/performing arts disability in Raynaud’s phenomenon participants (n = 37).

Quick-DASH ^†^: Sports/Performing Arts Disability (r^2 ‡^ = 0.275)						
Independent Variables	B ^§^	95% CI ^||^		β ^¶^	SE ^#^	*p*-Value
		Upper Limit	Lower Limit			
ROM ^††^ index finger extension	−1.530	−0.242	−2.817	−0.356	0.634	0.021 *
Lateral pinch strength	−6.095	−0.755	−11.43	−0.342	2.627	0.026 *

* *p* < 0.05; ^†^ Quick-DASH: Shortened Disability of the Arm, Shoulder and Hand questionnaire; ^‡^ r^2^: regression coefficient of determination; ^§^ B: regression coefficient; ^||^ CI: confidence interval; ^¶^ β: adjusted coefficient from multiple linear regression analysis; ^#^ SE: coefficient standard error; ^††^ ROM: range of motion.

**Table 6 diagnostics-14-00093-t006:** Final multiple regression model of predictive associated factors to work disability in RP participants (*n* = 37).

Quick-DASH ^†^: Work Disability (r^2 ‡^ = 0.418)						
Independent Variables	B ^§^	95% CI ^||^		β ^¶^	SE ^#^	*p*-Value
		Upper Limit	Lower Limit			
Temperature post-CST ^††^	−0.419	2.239	−3.078	−0.063	1.305	0.750
Oxygen Saturation	−15.532	−0.696	−30.368	−0.416	7.283	0.041 *
ROM ^‡‡^ index finger extension	−1.319	−0.011	−2.627	0.642	−0.297	0.048 *
ROM thumb flexion	−0.791	0.466	−2.048	0.617	−0.199	0.209

* *p* < 0.05; ^†^ Quick-DASH: Shortened Disability of the Arm, Shoulder and Hand questionnaire; ^‡^ r^2^: regression coefficient of determination; ^§^ B: regression coefficient; ^||^ CI: confidence interval; ^¶^ β: adjusted coefficient from multiple linear regression analysis; ^#^ SE: coefficient standard error; ^††^ CST: Cold Stress Test; ^‡‡^ ROM: range of motion.

## Data Availability

The data presented in this study are available on request from the corresponding author.

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
