# Peer review of "Analysis of Hand Function, Upper Limb Disability, and Its Relationship with Peripheral Vascular Alterations in Raynaud’s Phenomenon"

_diagnostics, 2023, doi:10.3390/diagnostics14010093_

Round 1
Reviewer 1 Report
Comments and Suggestions for Authors
This study provides an interesting perspective, but raises several concerns
(1) Please provide details regarding the underlying disease of SRP, especially the presence or absence of systemic scleroderma.
Norimatsu Y, Yoshizaki A, Kabeya Y, Fukasawa T, Omatsu J, Fukayama M, Kuzumi A, Ebata S, Yoshizaki-Ogawa A, Asano Y, Ichimura H, Yonezawa S, Nakano H, Sato S. Expert-Level Distinction of Systemic Sclerosis from Hand Photographs Using Deep Convolutional Neural Networks. J Invest Dermatol. 2021 Oct;141(10):2536-2539. doi: 10.1016/j.jid.2021.03.020. Epub 2021 Apr 7. PMID: 33836180.
(2) It is possible that the study was performed only in patients with mild SRP as a result of inclusion of patients with ulcers, etc. in the exclusion criteria.
This should be added to the "limitation" section.
(3) Did the participants have no history of treatment? Please specify whether there is any history of treatment that may affect blood flow.
Author Response
DIAGNOSTICS
Manuscript ID: diagnostics-2750191
Title: "Analysis of hand function, upper limb disability and its relationship with peripheral vascular alterations in Raynaud's phenomenon”.
REVIEWER 1:
This study provides an interesting perspective, but raises several concerns
Response: We would like to thank to reviewer for taking the time to review our manuscript and consider that our work provides an interesting perspective about Raynaud's phenomenon. Please find the detailed responses below and the corresponding revisions highlighted in the resubmitted manuscript file. We have highlighted in yellow colour all the changes we have made in the text
(1) Please provide details regarding the underlying disease of SRP, especially the presence or absence of systemic scleroderma.
Norimatsu Y, Yoshizaki A, Kabeya Y, Fukasawa T, Omatsu J, Fukayama M, Kuzumi A, Ebata S, Yoshizaki-Ogawa A, Asano Y, Ichimura H, Yonezawa S, Nakano H, Sato S. Expert-Level Distinction of Systemic Sclerosis from Hand Photographs Using Deep Convolutional Neural Networks. J Invest Dermatol. 2021 Oct;141(10):2536-2539. doi: 10.1016/j.jid.2021.03.020. Epub 2021 Apr 7. PMID: 33836180.
- Response: We appreciate your observation. As we mentioned in the introduction section, the secondary form of Raynaud's phenomenon is associated with an underlying disease, usually autoimmune diseases and connective tissue disorders. Within these disorders, Raynaud's phenomenon is present in 90% of patients with systemic sclerosis (Bech et al., 2017). This information is also supported by the updated reference that you have suggested us where it mentions that “Systemic Sclerosis presents with characteristic skin manifestations such as Raynaud's phenomenon and skin sclerosis from early onset” (Norimatsu et al., 2021), we have rewritten this information and included the reference in our manuscript to clarity. Page 2, lines 45-50.
“The secondary form of RP is associated with an underlying disease, usually autoimmune diseases and connective tissue disorders. Within these disorders, PR is a characteristic early-onset cutaneous manifestation that occurs in patients with systemic scleroderma (Norimatsu et al., 2021) and appears in up to 90 % of these patients, between 10-45 % with systemic lupus erythematosus, in 33 % with Sjögren's syndrome and in 20 % with dermatomyo-sitis/polymyositis (Belch et al., 2017).”
- References that support this information:
- Norimatsu,Y.; Yoshizaki, A.; Kabeya, Y.; et al. Expert-Level Distinction of Systemic Sclerosis from Hand Photographs Using Deep Convolutional Neural Networks. J Invest Dermatol, 2021,10, 2536-2539. doi: 10.1016/j.jid.2021.03.020
- Belch, J.; Carlizza, A.; Carpentier, P. H.; Constans, J.; Khan, F.; Wautrecht, J.C.; et al. ESVM guidelines-the diagnosis and management of Raynaud’s phenomenon. Vasa, 2017, 46, 413–423. doi: 10.1024/0301-1526/a000661
Moreover, all of the participants in our pilot study that were diagnosed with secondary Raynaud’s phenomenon had systemic scleroderma. So, all of the participants had Secondary Raynaud’s phenomenon to systemic scleroderma. We have added this relevant information in the results section and in the limitations section to clarity. Page 5, lines 200-201; page 18, lines 400-404.
“A total sample of 57 subjects with a mean (SD) age of 41.7 (15.5) years was recruited for this study: eighteen with PRP (72.2% females), nineteen with SRP to systemic scleroderma (78.9% females) and twenty healthy controls (80% females)”.
“Thirdly, we have only included in our study patients with SRP due to systemic sclero-derma with mild involvement since were excluded patients who presented scars, ulcers, gangrene or bites in the area to be examined, so are necessary future studies that include patients with SRP due to other pathologies and in different phases of involvement”.
(2) It is possible that the study was performed only in patients with mild SRP as a result of inclusion of patients with ulcers, etc. in the exclusion criteria.
This should be added to the "limitation" section.
- Response: We appreciate this observation. We agree with this comment, we have added this information in the limitations to highlight this important idea that you have suggested as you can see on page 18, lines 400-404.
“Thirdly, we have only included in our study patients with SRP due to systemic sclero-derma with mild involvement since were excluded patients who presented scars, ulcers, gangrene or bites in the area to be examined, so are necessary future studies that include patients with SRP due to other pathologies and in different phases of involvement”.
(3) Did the participants have no history of treatment? Please specify whether there is any history of treatment that may affect blood flow.
- Response: We appreciate this comment. The participants in our study maintained their usual pharmacological treatment which was based mainly on analgesic, nonsteroidal anti-inflammatory, vasodilatory, antidepressants and insulin drugs. We asked participants in our study about their current pharmacologic treatment at the time of the evaluation and recorded this information. We have added this information in the methodology section. Page 3, line 118.
“Firstly, participants were asked for their sociodemographic and clinical data, such as age, sex, hand dominance, current pharmacologic treatment…”
- We have also added the data that we have obtained about the participants' medical treatment at the time of the study in the results section. Page 5, lines 202-204.
“Regarding the pharmacological treatment of the participants in the study, it consisted mainly of analgesic drugs 21 (36.8%); non-steroidal anti-inflammatory drugs 16 (28.1%), vasodilators 8 (14%), antidepressants 7 (12.3%) and insulin 2 (3.5%)”.
Reviewer 2 Report
Comments and Suggestions for Authors
I read with interest the study by Rosa Mª Tapia-Haro et al. They reported RP has a disabling effect on upper extremities and practice of activities. In addition, they found disability in Raynaud seems to be more related with hand´s mobility and strength impairment than vascular injury. Their results are reliable, constructive, and valuble. This article deserves to be published.
Author Response
DIAGNOSTICS
Manuscript ID: diagnostics-2750191
Title: "Analysis of hand function, upper limb disability and its relationship with peripheral vascular alterations in Raynaud's phenomenon”.
REVIEWER 2:
- I read with interest the study by Rosa Mª Tapia-Haro et al. They reported RP has a disabling effect on upper extremities and practice of activities. In addition, they found disability in Raynaud seems to be more related with hand´s mobility and strength impairment than vascular injury. Their results are reliable, constructive, and This article deserves to be published.
Response: We woul like to thank the reviewer for his comments and for considering that our research is interesting, reliable, constructive and valuable, and that our article deserves to be published. Thank you very much